# Developmental Profile of Children with Autism Spectrum Disorder Versus Social Communication Disorder: A Pilot Study

**DOI:** 10.3390/children11101241

**Published:** 2024-10-15

**Authors:** Clara Dame, Marine Viellard, Sara-Nora Elissalde, Hugo Pergeline, Pauline Grandgeorge, Laure-Anne Garie, Federico Solla, Sonia De Martino, Elodie Avenel, Xavier Salle-Collemiche, Arnaud Fernandez, François Poinso, Elisabeth Jouve, Jokthan Guivarch

**Affiliations:** 1Medical School, Aix-Marseille University, 13385 Marseille, France; clara.dame@ap-hm.fr (C.D.); hugo.pergeline@ap-hm.fr (H.P.); francois.poinso@ap-hm.fr (F.P.); jokthan.guivarch@ap-hm.fr (J.G.); 2Department of Child Psychiatry, Sainte Marguerite Hospital, Assistance Publique des Hôpitaux de Marseille (APHM), 13009 Marseille, France; marine.viellard@ap-hm.fr (M.V.); sara-nora.elissalde@ap-hm.fr (S.-N.E.); pauline.grandgeorge@ap-hm.fr (P.G.); sonia.demartino@ap-hm.fr (S.D.M.); elodie.avenel@ap-hm.fr (E.A.); 3Speech-Language Pathology Department (École d’Orthophonie de Marseille), Aix-Marseille University, 13385 Marseille, France; anne-laure.garie@univ-amu.fr; 4Pediatric Surgery Department, Pediatric Hospitals of Nice CHU-Lenval (HPNCL), 57 Bd Californie, 06200 Nice, France; 5Department of Child Psychiatry, CMPI, Secteur 3, Centre Hospitalier des Pays de Morlaix, 15 Rue de Kersaint Gilly, 29600 Morlaix, France; xsalle@ch-morlaix.fr; 6SUPEA (University Department of Child and Adolescent Psychiatry), Expert Center for Pediatric Psychotrauma (CE2P) Pediatric Hospitals of Nice CHU-Lenval (HPNCL), 06200 Nice, France; arnaud.fernandez@hpu.lenval.com; 7EA CoBTek, UCA (University Côte d’Azur), 06100 Nice, France; 8Institut de Neurosciences de la Timone, UMR 7289, CNRS, Aix-Marseille University, 13005 Marseille, France; 9Public Health Department, Aix-Marseille University, 13005 Marseille, France; elisabeth.jouve@ap-hm.fr; 10Service Evaluation Médicale, Conception Hospital, Assistance Publique Hôpitaux de Marseille (APHM), 13005 Marseille, France

**Keywords:** Autism Spectrum Disorder, Social Communication Disorder, Diagnostic and Statistical Manual of Mental Disorders, comparative study, pragmatics language, Vineland Adaptive Behavior Scales, Autism Diagnostic Observation Schedule, Short Sensory Profile, Bishop’s Children’s Communication Checklist, Wechsler Intelligence Scale for Children IV

## Abstract

Background: Social Communication Disorder (SCD), introduced in the DSM-5, is distinguished from Autism Spectrum Disorder (ASD) by the absence of restricted and repetitive behaviors or interests (RRBIs). Aim: To compare the adaptive, sensory, communication, and cognitive profiles of children with ASD and SCD. Methods: The assessments of nine children with SCD and ten with ASD were compared with either Fisher’s Exact Test or the Mann–Whitney Test. Assessments included the Vineland Adaptive Behavior Scales, the Autism Diagnostic Observation Schedule (ADOS), the Short Sensory Profile, Bishop’s Children’s Communication Checklist, a pragmatics evaluation, and the Wechsler Intelligence Scale for Children IV. Results: The total ADOS score and the second subtotal “Restricted and Repetitive Behaviors” were significantly higher (*p* = 0.022) in the ASD group than in the SCD group. The Vineland standard score for the “Socialization” domain was significantly lower (*p* = 0.037) in the ASD group (mean: 51 +/− 19) than in the SCD group (mean: 80 +/− 28). The working memory index score was also significantly lower (*p* = 0.013) in the ASD group compared to the SCD group. Conclusions: While ASD and SCD share similarities in communication and pragmatic difficulties, some distinctions have been identified, e.g. in executive functioning and the impact on socialization, which may be linked to the absence of RRBIs in SCD. These findings highlight the challenges posed by this nosographic separation during diagnostic evaluations due to the scarcity of discriminative tools.

## 1. Introduction

The publication of the Diagnostic and Statistical Manual of Mental Disorders, Fifth Edition (DSM-5) allowed for the integration of neurodevelopmental disorders within our classifications. It modified the criteria for autism, which is now included in Autism Spectrum Disorder (ASD), and identified a new diagnostic category: Social Communication Disorder (SCD), classified among communication disorders [1].

### 1.1. Diagnostic Criteria for SCD and ASD

The diagnostic criteria for SCD in the DSM-5 are: “persistent difficulties in the social use of verbal and nonverbal communication in the following: -deficits in communication for social purposes-impairment of the ability to change communication to match context or needs of the listener-difficulties following rules for conversation or storytelling-difficulties understanding what is not explicitly stated and nonliteral or ambiguous meanings of language” [1].

SCD is a primary deficit in pragmatic language. Pragmatics involves the appropriate use of nonverbal (such as facial expression), paraverbal (such as intonation) and verbal communication, enabling adaptation according to the interlocutor and the social context [2]. Pragmatics is essential for effective communication in daily conversational exchanges, influencing both receptive and expressive aspects, and thereby playing a crucial role in social interactions [3]. 

In addition to the new diagnostic category represented by SCD, the criteria for ASD diagnosis were reorganized into two main dimensions: a social dimension, including deficits in communication and social relationships (criterion A); and a non-social dimension, including restrictive and repetitive behaviors and interests (RRBIs) (criterion B). Other diagnoses, such as Asperger’s syndrome, have been consolidated into the single entity of ASD in the DSM-5, provided the symptomatic dyad is met. The DSM-5’s classification of ASD considers severity levels and identifies additional characteristics (such as intellectual impairment, language developmental delays, catatonia, and co-occurrence with genetic, medical, or environmental factors). This approach allows for a detailed description of the diverse presentations within ASD and facilitates the inclusion of the full spectrum of symptoms associated with this disorder.

### 1.2. Very Similar Diagnoses

Although they are articulated differently, the diagnostic criteria for SCD closely resemble those of the social dimension of ASD, particularly concerning deficits in verbal and non-verbal communication. It is noteworthy that deficits in pragmatics, especially related to deficiencies in inferential abilities, are crucial for identifying SCD. In comparison, the diagnostic criteria for ASD focus on impairments in social interactions and socio-emotional reciprocity. Individuals with SCD may have a typical social drive [4]. Simms and Jin [5] describe individuals with SCD as having well-developed imaginative capacities, participating in pretend play in contrast to children with ASD. 

The main difference between SCD and ASD is the absence of RRBIs in SCD. This criterion includes repetitive use of objects, behavioral or verbal rituals, compulsions, inflexible adherence to routines, sensory features, restricted interests or unusual preoccupations, and repetitive or stereotyped movements. In ASD, these elements may sometimes only be present during the early developmental period, which complicates the differential diagnosis and requires a detailed history. Furthermore, repetitive or compulsive behaviors can be found in typically developing children [5,6], which raises questions about a pathological threshold for RRBIs. The DSM-5 requires the presence of two out of four items to meet Criterion B for ASD. From this premise, a question arises: can children with present but insufficient symptoms for an ASD diagnosis be diagnosed with SCD? Recent literature shows that the distinction between the absence of RRBIs in SCD and their presence in ASD is not well-defined [7,8].

While the global prevalence of autism is known to be around 1%, with a sex ratio of four boys to one girl [5,9], it is more difficult to establish the prevalence of SCD due to the limited number of studies conducted. Kim et al. [10] compared the diagnostic criteria of DSM-IV and DSM-5 in a school-aged population in South Korea, and calculated a prevalence of SCD of around 0.5%. Jo Saul et al. [11], using Bishop’s Children’s Communication Checklist (CCC), reported a prevalence between 0 and 1.3% for social-pragmatic deficits in children aged 5–6 years without autistic symptoms and without structural language difficulties. Ellis Weismer et al. [8] found a similar sex ratio to ASD, with four boys for every one girl, among 13–14-year-olds who met the SCD criteria via the CCC.

With the DSM-IV, children with communication and social interaction difficulties but without RRBIs could be classified under the category of Pervasive Developmental Disorder Not Otherwise Specified (PDD-NOS) [12]. This diagnostic category allowed for the inclusion of patients with an incomplete presentation of autism, enabling them to access similar care and educational support as children with a specified pervasive developmental disorder such as autism. In the South Korean study by Kim et al. [10], 71% of cases diagnosed with PDD-NOS according to the DSM-IV met the criteria for ASD in the DSM-5, and 22% of PDD-NOS cases met the criteria for SCD. It is also noteworthy that 1% of autism cases according to the DSM-IV did not meet the criteria for ASD, but did meet the criteria for SCD, in the DSM-5. Mandy et al. [13], out of the 88 cases diagnosed with SCD, found 26 cases (29.5%) who met the criteria for PDD-NOS under the DSM-IV, 24 cases (27.3%) who met the criteria for autism or Asperger’s syndrome, and 38 cases (43.2%) who did not meet any PDD criteria. Children with SCD do not uniformly correspond to those who would have been diagnosed with PDD-NOS with the DSM-IV.

### 1.3. Absence of a Gold Standard for SCD

One of the challenges encountered in studies on SCD, aside from the small study population, is the absence of a gold standard: there is no validated tool for the diagnosis of SCD. The diagnostic criteria for SCD primarily focus on difficulties related to pragmatic language. However, no standardized tool exists to fully evaluate the broad and complex field of pragmatics [14]. The Autism Diagnostic Interview-Revised (ADI-R) and the Autism Diagnostic Observation Schedule (ADOS) are recognized as reliable tools for the diagnosis of ASD, with good specificity. Foley-Nicpon et al. [15] relied on the study by Mazefsky et al. [16] to compare the diagnosis of ASD (DSM-5) and autism (DSM-IV) using items from the ADI-R and ADOS. They also aimed to examine the correspondence of items from these two tools with the four DSM-5 criteria for SCD in children who were not diagnosed with ASD under the DSM-5. However, they found no items categorized as identifying the fourth criterion of SCD (the ability to understand implicit language or figurative or ambiguous expressions), whereas all four criteria must be present to make the diagnosis. The diagnostic tools typically used for ASD do not seem sufficient for a precise and comprehensive evaluation of SCD, according to the DSM-5 definition. Some studies used Bishop’s Children’s Communication Checklist [8,17], which provides a score assessing various domains of pragmatics, to identify children meeting the criteria for SCD. Nevertheless, the CCC is a scale completed by caregivers, which may reduce its reliability. Further studies are needed to evaluate the relevance of these tools for diagnosing children with SCD.

### 1.4. Study Question and Hypotheses

The question arises whether SCD corresponds to a distinct clinical picture, previously described in the literature as Pragmatic Language Impairment (PLI), notably by Bishop [18,19], or as a less severe form of ASD [8,20]. The diagnosis of SCD, its relevance, and clinical utility continue to be debated in the literature, particularly regarding its distinction from ASD in terms of developmental, biological, or prognostic aspects. This distinction suggests a need for specific recommendations to diminish the risk of reduced recognition of the disorder and limited access to public health programs, as well as educational and professional support and accommodations [21].

Based on our clinical experience, we hypothesized that children with SCD differed from those with ASD, particularly in terms of social skills. We also hypothesized that children with SCD would score below the ASD threshold at ADOS-2.

We therefore sought to determine whether and how children diagnosed with SCD differed from those diagnosed with ASD, through various assessments classically conducted in child psychiatry departments. Highlighting differences in developmental profiles would thus make it possible to propose tailored therapeutic recommendations for children with SCD.

### 1.5. Objective

The main objective of this pilot study was to compare the developmental profiles of children diagnosed with ASD versus those diagnosed with SCD. The cognitive, adaptive, sensory, and communication profiles, and their imaginative capacities, were examined. We expected to find more socialization deficits in children with ASD.

## 2. Materials and Methods

### 2.1. Participants

The recruitment was conducted based on outpatients aged 8 and above who underwent a diagnostic evaluation in the child psychiatry department at Sainte Marguerite University Hospital (Marseille, France). 

We retrospectively included children with social communication disorder for the SCD group and autism spectrum disorder for the ASD group, according to DSM-5 criteria, based on the medical files of patients assessed between 1 January 2016, and 31 December 2018. The first 10 patients who received a diagnosis and met the inclusion and non-inclusion criteria were included in each group, resulting in a total of 20 participants. 

This inclusion period and sample size were determined by considering the prevalence of SCD, the observation that this diagnosis was infrequently encountered in psychiatric consultations, and the design of this preliminary study. Psychological and pedagogical practice shows that not all cases of children and adolescents with SCD symptoms are registered in mental health clinics [11].

Children with epilepsy, intellectual disability, insufficient oral language for communication, or whose parents did not have sufficient proficiency in French were not included in the study. Children were excluded if more than two assessments among those analyzed in our study had not been conducted during the diagnostic evaluation, or if consent for participation was withdrawn.

In order to study the initial comparability of the two patient groups, sociodemographic characteristics recorded during medical consultations were collected and compared. These data included the child’s age and sex, the number of siblings and their birth order in the family, the mode of parental authority (whether the parents are together or separated), the parents’ socio-professional category, as well as the child’s and siblings’ medical history, e.g., prematurity or perinatal history, the child’s comorbidities, and any disorders among siblings. Information on the child’s schooling (ordinary or specialized class) was also collected.

### 2.2. Measurements

Various assessments were conducted, as follows.

#### 2.2.1. Autism Diagnostic Observation Schedule 2 (ADOS-2) [22]

The ADOS is a semi-structured observational assessment for confirming ASD diagnosis, adapted in French by B. Rogé et al [23]. The patient is placed in social situations where they must interact, including tasks involving imaginative play, use of materials, expression of emotions, and conversation. Two subtotals are calculated corresponding to the two criteria for ASD from the DSM-5: qualitative impairment in social interactions and communication on the one hand, and repetitive or restricted interests and activities on the other hand. An algorithm gives a total score, and elevated scores place an individual in the autism spectrum or autism diagnostic range. Additionally, during the ADOS, imaginative capacity is assessed through one criterion, which evaluates pretend play and forms of creativity in play and conversation. There are different ADOS modules depending on the patient’s age and language level; in this study, Module 3 was used for all patients. 

A calibrated severity score provides information about ASD severity with relative independence from individual characteristics such as age and verbal IQ.

#### 2.2.2. Vineland Adaptive Behavior Scale II (VABS-II) [24] 

The VABS-II or Vineland-II is a scale that allows for the evaluation of a child’s level of autonomy and adaptation by determining a developmental age. The assessment uses a semi-structured interview of caregivers to obtain information and scores in three to four domains depending on the child’s age:-Communication, including the subdomains: receptive, expressive, and written.-Daily living skills, including the subdomains: personal, domestic, and community.-Socialization, including the subdomains: interpersonal relationships, play and leisure, and coping skills.-Motor skills, assessed only for children under 7 years old, were not used in this study.

After algorithmic transformation, the total score allows for the calculation of an adaptive quotient (AQ). This scale, in addition to the psychometric assessment, is useful for evaluating the level of functioning of any developmental disorder. The second version of this scale, referencing the DSM-5, was developed in 2015 and has been adapted into French.

#### 2.2.3. Short Sensory Profil (SSP) [25] 

The SSP is a reliable and valid parent report measure of behaviors associated with atypical responses to sensory stimuli (always, frequently, sometimes, rarely, never), translated into French [26]. This short version includes 38 items divided into seven sections: Tactile Sensitivity, Taste/Smell Sensitivity, Movement Sensitivity, Underresponsive/Seeks Sensation, Auditory Filtering, Low Energy/Weak, and Visual/Auditory Sensitivity. Scores are compared to standardized data from 561 typically-developing children. A definite difference indicates scores greater than two standard deviations from the mean for typically developing children in the standardization sample, whereas a probable difference indicates scores greater than one and less than two standard deviations from the mean. 

#### 2.2.4. Bishop’s Children’s Communication Checklist 2 (CCC-2) [27]

The CCC-2 is a questionnaire that measures children’s structural and pragmatic language abilities, completed by a rater (parent or teacher) who has been familiar with the child. Raters make a frequency judgment for each of the 70 items. This checklist includes nine subscales, five of which relate to pragmatics and allow for the calculation of a pragmatic component score: initiation, coherence, stereotyped language, context, and conversational rapport.

This scale has one standardization for children with language impairment and another standardization for children without language impairment. For each child in the study, the standard deviation from the mean was calculated according to the presence or absence of language difficulties. Language impairment was assessed by speech therapists, using clinical examination and assessments such as the ELO or ECL by A. Khomsi, I. Nanty, F. Pasquet, and A. Parbeau-Guéno.

#### 2.2.5. Evaluation of Pragmatics

Pragmatic impairments were assessed during a clinical interview by speech therapists from the department. They used the same observation and evaluation framework, called in French “Trame pour l’Observation et le Soin Structurés de la Pragmatique du Langage” (TOSS-PL, Framework for structured observation and care of language pragmatics) [28], which is detailed in eight domains:-Non-verbal-Paraverbal-Function of communication and language-Management of conversational exchange-Adaptation and reciprocity of discourse-Organization of discourse-Enunciation and modes of language investment-Semantic elements with high pragmatic impact

The assessment involves various steps that include the use of standardized tools, such as a narrative task and/or narrative comprehension norm-referenced tests focused on pragmatics and periods of spontaneous conversation. This framework provides a (non-standardized) scoring system based on these eight domains, allowing for the identification of therapeutic strategies according to the intensity of any difficulties and/or specificities identified. This scoring system was used in our study to identify the most affected domains.

#### 2.2.6. Wechsler Intelligence Scale for Children (WISC) [29]

The WISC is a standardized test of intellectual ability, used for children aged 6–16 years. The WISC-IV provides four indices: -Verbal Comprehension Index (VCI) using 3 subtests: similarities, vocabulary, and comprehension. These subtests measure verbal knowledge and verbal concepts.-Perceptual Reasoning Index (PRI) using 3 subtests: block design, picture concepts, and matrix reasoning. These subtests measure logical reasoning, visuospatial processing and visuomotor integration.-Working Memory Index (WMI) using 2 subtests: digit span and letter-number sequencing. The tasks related to verbal working memory provide information on auditory-verbal short-term memory.-Processing Speed Index (PSI) using 2 subtests: coding and symbol search. These subtests measure the ability to quickly process visual information and visual attention.

### 2.3. Evalaution Procedure 

The developmental profile of the children in each group was evaluated using the assessments usually conducted in the department during a diagnostic evaluation.

Thus, the adaptive profile was assessed with the VABS-II scale completed by doctors or psychologists. The sensory profile was determined through a parental questionnaire using the Short Sensory Profile conducted by the psychometricians. The communication profile was evaluated both during the ADOS-2 by trained psychologists and through assessments conducted by the two speech therapists participating in the study, who performed a language evaluation (if not previously done), CCC-2, and a pragmatic assessment. The capacity for imagination was assessed during the ADOS. The cognitive profile was evaluated using the WISC-IV administered by psychologists.

### 2.4. Analysis

Each child was assigned a code based on the chronological order of their inclusion and the initials of their first and last names. An Excel file was created to collect the data. 

Qualitative variables, expressed in frequencies and percentages, were compared using Fisher’s Exact Test, due to small sample sizes. Quantitative variables were compared using the Mann–Whitney Test, based on mean ranks. Statistical tests were exploratory. The significance level was set at 0.05 in a two-sided situation. Statistical analyses were performed using SPSS software (version 20). 

## 3. Results

### 3.1. Description of the Population

Twenty patients met the inclusion criteria. Among them, one patient in the SCD group was excluded because more than two assessments analyzed in this study were not conducted during the diagnostic evaluation (Figure 1). 

The various characteristics of the participants were compared between groups, and no significant differences were found (Table 1).

However, it is noteworthy that no girls were included in the SCD group compared to two in the ASD group. Additionally, the children in the SCD group were between 9 and 15 years old (median age of 12), while the children in the ASD group were between 8 and 14 years old (median age of 9.5) (*p* = 0.082). Regarding interventions, four children in the SCD group (44.4%) were receiving speech therapy compared to nine children in the ASD group (90%) at the time of inclusion (*p* = 0.057). 

### 3.2. Principal Results

The ADOS-2 was conducted with eight patients in the SCD group and ten patients in the ASD group. The second subtotal “Restricted and Repetitive Behaviors” (RRB) was significantly higher (*p* = 0.018) in the ASD group (mean: 1.8 +/− 1) than in the SCD group (mean: 0.5 +/− 0.8). Nine out of 10 children in the ASD group (90%) had an RRB score of 1 or higher, compared to only three out of eight children in the SCD group (37.5%) (*p* = 0.043). The total ADOS score was also significantly higher (*p* = 0.022) in the ASD group (mean: 8.8 +/− 2.6) than in the SCD group (mean: 4.9 +/− 3.3). 

In the ASD group, according to the ADOS 2 calibrated severity score, three patients had mild autistic symptoms, six had moderate symptoms and only one had severe symptoms.

No significant difference was found regarding the evaluation of imagination during the ADOS (Table 2).

The VABS-II was completed by eight children in the SCD group and ten children in the ASD group. The standard score for the “Socialization” domain was significantly lower (*p* = 0.037) in the ASD group (mean: 51.4 +/− 19.1) than in the SCD group (mean: 79.9 +/− 28.1). No significant differences were found between the two groups for the other domain standard scores or the “Adaptive Behavior Composite” score (Figure 2).

The SSP was completed by eight patients in each group. No significant differences were found regarding the presence of definite or probable sensory differences, or the type of sensory sensitivity (Table 3).

The complete speech therapy assessment was conducted with nine patients in the SCD group and nine patients in the ASD group. Two patients in the SCD group had language difficulties compared to four patients in the ASD group. The pragmatic component score (CCC-2) was significantly lower (*p* = 0.03) in the ASD group (mean: 130.8 +/− 5.8) than in the SCD group (mean: 137.8 +/− 7.5), but the difference between the standard deviations of each group was not significant. No differences were found in the evaluation of pragmatics (Table 4).

All the included patients completed a WISC. The WMI score was significantly lower (*p* = 0.013) in the ASD group compared to the SCD group. The standard score on the Digit Span subtest, which is used to calculate the WMI, was also significantly lower (*p* = 0.024) in the ASD group (mean: 5.9 +/− 1.9) than in the SCD group (mean: 8.6 +/− 2.7) (Table 5).

## 4. Discussion

This study compared the developmental profiles of 19 children over the age of 8 diagnosed with SCD or ASD through various assessments conducted during the diagnostic evaluation: WISC, ADOS-2, VABS-II, Short Sensory Profile, CCC-2, and an evaluation of TOSS-PL. The results show significant differences in certain domains, which could indicate different developmental profiles between these two disorders.

### 4.1. Discussion of the Population 

The two groups were comparable in terms of sociodemographic criteria. The low proportion of girls (two girls in the ASD group and none in the SCD group) can be explained by the known and estimated sex ratios of these disorders. Numerous studies describe possible differences in clinical presentation based on the sex of individuals with psychiatric disorders such as ASD, which can lead to a later diagnosis for girls [30]. Another possible explanation for the absence of girls in the SCD group might be the increased difficulty in diagnosing girls. 

The high rate of comorbidities in the ASD group is consistent with current knowledge [1], and a similar rate could also be expected in the SCD group. However, despite their relatively comparable frequency in both groups, the presence of associated comorbidities such as Attention Deficit Hyperactivity Disorders (ADHD) or learning disorders may influence the results.

### 4.2. Discussion of the Outcomes

On the ADOS, the RRB subtotal was higher in the ASD group, which is consistent with the diagnostic criteria for ASD, based on a symptom dyad including RRBIs. Only one child in the ASD group had a score of 0 on this subtotal. RRBIs may only be present in ASD patients during an early developmental period. This was the case for the child diagnosed with ASD who scored 0 in the ADOS RRBIs subscale, but scored significantly for RRBIs in the ADI-R, which evaluates early developmental periods. Indeed, the ADI-R detected the presence of restricted interests and motor stereotypies, and the sensory profile also found definite sensitivities. Moreover, it is worth noting that this child is female, and as mentioned earlier, the clinical presentation may differ, potentially influencing the ADOS assessment. In this regard, Rea et al. [31] found significantly lower scores in both ADOS subtotals and the total score for girls compared to boys. Mandy et al. [32] found a difference only in the RRBIs, both on the ADOS and the parental interview (3Di). Conversely, in our SCD group, three children scored 1 on the RRBIs criterion. For these three children, RRBIs were lower than criterion B for ASD diagnosis during family interviews, consultations, or on the ADI-R. Despite the significant difference found in our study, the ADOS does not seem to be sufficient to distinguish certain ASDs from SCD due to the variability in symptom intensity and the limited observation time provided by this brief and standardized assessment. The question remains regarding the diagnostic classification of children showing RRBIs below the pathological level.

Imagination was assessed by a single item rating during the ADOS, and our results do not confirm the hypothesis that pretend play is more developed in patients with SCD compared to those with ASD. Future research will be necessary to evaluate imagination more specifically, but there are few specialized tools available for this purpose.

Regarding the VABS-II, our population of children with ASD had an adaptive profile similar to that described in the manual for verbal children with ASD without ID, and whose socialization is affected more than communication [24]. Studies have described this “typical autism profile” using the first version of the Vineland [33,34]. A different profile was found for children with SCD, with better adaptive abilities in the area of socialization. These abilities could be explained by the absence of RRBIs, which have the potential to interfere with social interactions [35] and cooperative play. This adaptive profile is consistent with the hypothesis that children with SCD have an inclination for interaction, good imaginative, and pretend play abilities.

On the SSP, no significant differences were found, even though sensory features are part of the RRB criterion. It is noteworthy that all children in the ASD group had probable or definite sensitivities, which is consistent with previous studies [36]. However, the most frequently reported types of sensitivities (hyporeactivity/sensation seeking, auditory filtering, tactile sensitivity, taste/smell sensitivity) partially differ from those found in our study (auditory filtering, taste/smell sensitivity, low energy/weakness), likely due to our small sample size [36]. Additionally, among the three children with definite sensitivities in the SCD group, two have comorbid ADHD and score in the “auditory filtering” and “movement sensitivity” sections. Some studies have shown that sensory features can also be observed in children diagnosed with ADHD. The latter score is significantly higher than typically developing children on the SSP [37]. 

Regarding the CCC-2, the pragmatic composite score was lower in children with ASD, but this difference was not observed in the standard deviation, which accounts for language difficulties (two in the SCD group versus four in the ASD group). In the evaluation of pragmatics in our study, the intensity or profile of pragmatic deficiencies does not seem to effectively distinguish between the two disorders. The similarity between the first criterion of ASD and the diagnostic criteria of SCD explains these results. We can observe that in both groups, there is inter-individual variability in the most impacted areas of pragmatics. Despite pronounced pragmatic impairments in the SCD group, the proportion of children receiving speech therapy was not equal (four patients in the SCD group compared to nine in the ASD group, *p* = 0.057). This difference, although not significant, could be explained by a milder clinical presentation or a lack of awareness about the disorder. 

The differences identified in the WISC concern the WMI, particularly the “Digit Span” subtest. These results suggest a different cognitive functioning between the two groups of patients, particularly poorer auditory working memory skills in the ASD group. Numerous studies have examined working memory deficiencies in patients with ASD, with sometimes contradictory results. In a 2020 study, Audras-Torrent and his team [38] examined the WISC profiles of 121 children with ASD without ID and found better results on tests involving visual working memory compared to those involving auditory working memory. Meanwhile, a meta-analysis [39] showed that visual working memory was more impaired than auditory working memory. The latter comprises one of the core elements of executive function, and executive dysfunctions in children with ASD are one of the theories explaining certain symptoms, such as rituals and cognitive rigidity. This difference could thus be consistent with the definition of SCD, in which these symptoms are excluded.

### 4.3. Strengths and Limitations

The main strength of this study is the use of various assessment tools that provide a comprehensive profile across different areas of development. These assessments can be divided into two categories: those based on parental reports and those based on professional observations, enabling an evaluation of children from multiple perspectives. To the authors’ knowledge, this is the first study comparing the profiles of children with SCD and ASD in terms of adaptive, sensory, cognitive, communication, and imagination aspects.

However, this study has several limitations. First, the small sample size, partly due to the prevalence of SCD, could limit the study’s relevance and reliability. Moreover, only 14 participants (seven from each group) completed all the assessments analyzed in the study. Second, no matching by age or sex was performed. Although no significant differences were found between the two groups, the absence of matching limits the comparisons. Third, we focused on verbal children with ASD without ID to allow for comparison with the SCD group, which means the results cannot be generalized to all children with ASD. 

### 4.4. Implications for Practice

Few studies have been conducted since the introduction of SCD in the DSM-5 [40]. Although there is substantial literature on pragmatic disorders such as PLI, our understanding of SCD remains limited. This study aimed to enhance our knowledge of this disorder. While ASD and SCD share similarities in communication and pragmatic difficulties, some distinctions have been identified, such as differences in executive functioning and the impact on socialization, which may be linked to the absence of RRBIs. These findings also highlight the challenges posed by this nosographic separation during diagnostic evaluations due to the lack of discriminative tools. Our preliminary results indicate different developmental profiles between children with ASD and those with SCD. Future studies with larger patient samples and long term follow up are needed to determine whether these differences establish SCD as a fundamentally distinct disorder from ASD in terms of diagnosis, prognosis, and care, to support the development of specific therapeutic recommendations.

## 5. Conclusions

Through a pilot study comparing the developmental profile of children with ASD versus those with SCD, some differences have been identified. In particular, it appears that in addition to the RRBIs present in the ASD group and not in the SCD group, socialization skills are less impaired in the SCD group. Moreover, the total score on the ADOS-2 is lower in SCD, on average below the significant cut-off score for SCD.

While these initial results are interesting, it is important to replicate the study to confirm them with a larger sample. In addition, it would enable us to compare cases of SCD with various forms of ASD depending on the level of intensity of the symptoms.

## Figures and Tables

**Figure 1 children-11-01241-f001:**
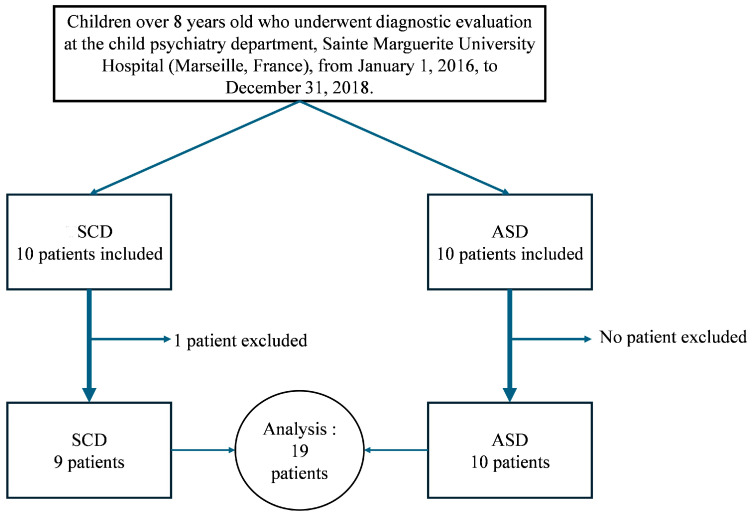
Flow chart.

**Figure 2 children-11-01241-f002:**
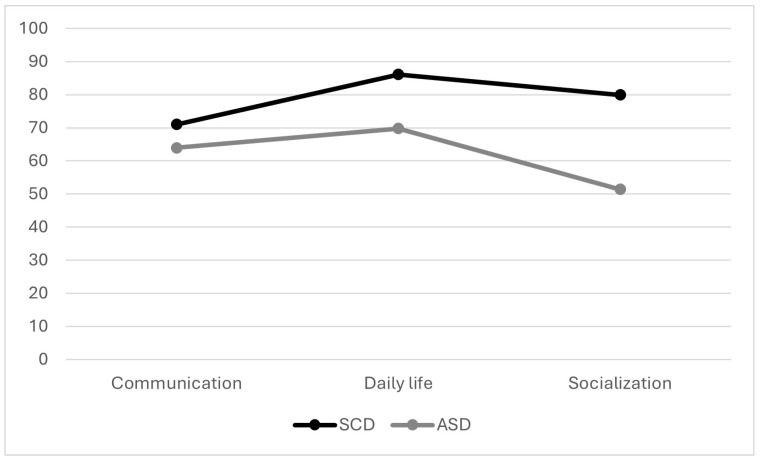
Adaptive profile based on mean standard scores across domains on the VABS-II.

**Table 1 children-11-01241-t001:** Characteristics of participants.

Variables	TCS *n* = 9	TSA *n* = 10
Gender		
Male *n* (%)	9 (100)	8 (80)
Female *n* (%)	0 (0)	2 (20)
Age (median (min; max))	12 (9; 15)	9.5 (8; 14)
Perinatal history *n* (%)	3 (33.3)	1 (10)
Comorbidities *n* (%)		
None	3 (33.3)	5 (50)
Specific learning disorder.	2 (22.2)	4 (40)
Attention Deficit Hyperactivity Disorder	2 (22.2)	3 (30)
Anxiety disorders	1 (11.1)	0 (0)
Tics	1 (11.1)	1 (10)
Siblings *n* (%)		
Presence of siblings	9 (100)	7 (70)
Eldest sibling	7 (77.8)	4 (57.2)
Presence of disorders among siblings (%)	1 (11.1)	1 (14.3)
Parents in a couple *n* (%)	9 (100)	8 (80)
Socio-professional category (PCS) (Mother/Father) *n* (%)		
Unemployed	2 (22.2)/0 (0)	1 (10)/0 (0)
1	0 (0)/0 (0)	0 (0)/0 (0)
2	0 (0)/3 (33.3)	3 (30)/1 (11.1)
3	2 (22.2)/4 (44.4)	0 (0)/0 (0)
4	4 (44.4)/1 (11.1)	1 (10)/3 (33.3)
5	1 (11)/0 (0)	5 (50)/3 (33.3)
6	0 (0)/1 (11)	0 (0)/2 (22.2)
Treatments *n* (%)		
Speech therapy	4 (44.4)	9 (90)
Psychomotor therapy	4 (44.4)	5 (50)
Individual psychological therapy	4 (44.4)	5 (50)
Group psychological therapy	1 (11.1)	1 (10)
Occupational therapy	3 (33.3)	2 (20)
Specialized educator	0 (0)	1 (10)
Psychopedagogy	4 (44.4)	1 (10)
Schooling in specialized class *n* (%)	1 (11.1)	3 (30)

PCS: classification of socio-professional categories in France. Psychopedagogy: day care hospital in partnership with the national education system.

**Table 2 children-11-01241-t002:** Scores and sub-scores on the ADOS-2.

Scores	SCD *n* = 8	ASD *n* = 10	*p*-Value
Social affect			
Median (min; max)	4.5 (0; 8)	7 (5; 9)	0.192
Mean ± SD	4.9 ± 3.2	7 ± 1.4	
Repetitive and restricted behavior or interest			
Median (min; max)	0 (0; 2)	1.5 (1; 3)	0.018 *
Mean ± SD	0.5 ± 0.8	1.8 ± 1.0	
Total ADOS-score			
Median (min; max)	5.5 (0; 8)	8.5 (6; 10)	0.022 *
Mean ± SD	4.9 ± 3.3	8.8 ± 2.6	
ADOS item Imagination/Creativity			
=0 *n* (%)	1 (12.5)	2 (20)	1
=1 *n* (%)	7 (87.5)	8 (83.3)	

* indicates *p* < 0.05.

**Table 3 children-11-01241-t003:** Types of sensitivities in the SSP.

Results	SCD *n* = 8	ASD *n* = 8	*p*-Value
Definite difference *n* (%)	3 (37.5)	5 (62.5)	0.619
Definite and/or probable difference (%)	7 (87.5)	8 (100)	1
Sensory section (definite and/or probable) *n* (%)			0.836
Tactile sensitivity	1 (14.3)	2 (25)	
Taste/smell sensitivity	1 (14.3)	4 (50)	
Movement sensitivity	2 (28.6)	3 (37.5)	
Underresponsive/seeks sensation	1 (14.3)	3 (37.5)	
Auditory filtering	6 (85.7)	7 (87.5)	
Low energy/weak	3 (42.9)	5 (62.5)	
Visual/auditory sensitivity	1 (14.3)	2 (25)	

**Table 4 children-11-01241-t004:** Speech-language assessment results (Oral language, CCC-2, pragmatics evaluation).

Results	SCD *n* = 9	ASD *n* = 9	*p*-Value
Language impairment *n* (%)	2 (22.2)	4 (44.4)	0.62
CCC-2 pragmatic component score (mean ± SD)	137.8 ± 7.5	130.8 ± 5.8	0.03 *
CCC-2 standard deviation from the mean (mean ± SD)	−2.03 ± 1.40	−2.36 ± 0.78	0.51
Pragmatics evaluation (mean ± SD)			
Non verbal	−2.4 ± 1	−2.1 ± 0.6	0.32
Para-verbal	−1.6 ± 1.4	−2.7 ± 0.7	0.78
Function of communication and language	−2 ± 0.8	−2.1 ± 0.9	0.93
Management of conversational exchange	−2.5 ± 0.8	−1.5 ± 1.6	0.24
Adaptation and reciprocity of discourse	−2.9 ± 0.4	−2.7 ± 0.8	0.78
Organization of discourse	−2.8 ± 0.7	−2.8 ± 0.8	0.89
Enunciation and modes of language investment	−0.4 ± 1.4	−1.7 ± 1.5	0.1
Semantic elements with high pragmatic impact	−0.9 ± 1.3	−1.1 ± 1.9	0.78

* indicates *p* < 0.05.

**Table 5 children-11-01241-t005:** Index scores on the WISC-IV.

Index	SCD *n* = 9	ASD *n* = 10	*p*-Value
VCI (mean ± SD)	107.1 ± 19.5	100.2 ± 8.5	0.682
PRI (mean ± SD)	102.4 ± 21	102.8 ± 16.7	0.806
WMI (mean ± SD)	91.9 ± 20.1	77.7 ± 9.4	0.013 *
PSI (mean ± SD)	89.4 ± 13.6	78.2 ± 10.7	0.071

* indicates *p* < 0.05.

## Data Availability

All data generated for this study are available from the corresponding author upon reasonable request. The data are not publicly available to protect the data privacy and restrict unauthorized use.

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
