# Peer review of "Developmental Profile of Children with Autism Spectrum Disorder Versus Social Communication Disorder: A Pilot Study"

_children, 2024, doi:10.3390/children11101241_

Round 1

Reviewer 1 Report

Comments and Suggestions for Authors

Psychological and pedagogical practice shows that not all cases of children and adolescents with SCD symptoms are registered in mental health clinics. Teachers and caregivers of these children are aware of this. The authors should mention this in the methodological section.

The number of subjects in both groups compared is relatively small. This means that the conclusions from the analysis of the results are of limited value. In addition, it should be remembered that ASD includes cases of disorders that differ significantly in terms of the picture of the symptoms and their intensity, which makes this a highly heterogeneous group. This means that caution is needed when making appropriate comparisons. It is good that the authors drew attention to the limitations of their research in the final part of the discussion of the results. Based on these comments, it seems appropriate for the authors to indicate that their study is a pilot study. I suggest that the authors should repeat the study on a larger population of children and adolescents with SCD and ASD. In such a study, the analysis of the results could be extended to include a comparison of cases with SCD with various forms of ASD depending on the levels of intensity of the symptoms.

The strength of the conducted research is the selection of research tools, adequate to the undertaken topic, as well as the method of conducting the analysis of the results. Another advantage is the possibility of using the obtained results when making a differential diagnosis in the practical activities of psychologists and psychiatrists. I support the publication of the reviewed article after taking into account the comments and suggestions listed above.

Author Response

Editor:

(I) Ensure all references are relevant to the content of the manuscript.

(II) Highlight any revisions to the manuscript, so editors and reviewers can

see any changes made.

(III) Provide a cover letter to respond to the reviewers’ comments and

explain, point by point, the details of the manuscript revisions.

(IV) If the reviewer(s) recommended references, critically analyze them to

ensure that their inclusion would enhance your manuscript. If you believe

these references are unnecessary, you should not include them.

(V) If you found it impossible to address certain comments in the review

reports, include an explanation in your appeal.

Authors’ responses

  1. All references are relevant to the content of the manuscript.
  2. We have highlighted all changes to the manuscript in yellow.
  • We have provided a cover letter to respond to the reviewers’ comments and have explained, point by point, the details of the manuscript revisions.
  1. Reviewers did not recommend references.
  2. We have responded to all reviewers' comments

Reviewer #1

Psychological and pedagogical practice shows that not all cases of children and adolescents with SCD symptoms are registered in mental health clinics. Teachers and caregivers of these children are aware of this. The authors should mention this in the methodological section.

Authors’ responses

As requested by reviewer 1, we have added the sentence “Psychological and pedagogical practice shows that not all cases of Children and adolescents with SCD symptoms are registered in mental health clinics” to the methods section (subsection “participants”). We have added reference 11 (already cited above).

In fact, the authors of this ecological study in schools, highlighted socio- pragmatic language disorders in children not previously treated in mental health clinics - and only some of whom met the diagnostic criteria for SCD - and the link with behavioral problems and poor academic skills.

The number of subjects in both groups compared is relatively small. This means that the conclusions from the analysis of the results are of limited value. In addition, it should be remembered that ASD includes cases of disorders that differ significantly in terms of the picture of the symptoms and their intensity, which makes this a highly heterogeneous group. This means that caution is needed when making appropriate comparisons. It is good that the authors drew attention to the limitations of their research in the final part of the discussion of the results. Based on these comments, it seems appropriate for the authors to indicate that their study is a pilot study.

Authors’ responses

The small size of our groups and the fact that we only included verbal children with no intellectual deficits limit the generalizability of our results, given the wide clinical variability of ASD.

We have made this clear in the limitations.

Following the reviewer's suggestion, we have indicated in the title and text that our study is a “pilot study”.

I suggest that the authors should repeat the study on a larger population of children and adolescents with SCD and ASD. In such a study, the analysis of the results could be extended to include a comparison of cases with SCD with various forms of ASD depending on the levels of intensity of the symptoms.

Authors’ responses

We have incorporated - in the conclusion section - the reviewer's suggestion to replicate the study with a larger number of subjects that will enable a comparison with different clinical forms of ASD. We have added a paragraph.

While these initial results are interesting, it is important to replicate the study to confirm them with a larger number of children. In addition, it would enable us to compare cases of SCD with various forms of ASD depending on the level of intensity of the symptoms. »

Reviewer #2

1) The Introduction is well-written, and provides important information, the goals are clear. However, the authors should add study questions or / and hypotheses. In addition, it would be helpful to add headings in the introduction.

Authors’ responses

At the request of reviewer 2, we have added our hypotheses at the end of the introduction.

We have also added headings in the introduction.

2) The sample size is small. This limits the possibility of generalizing the results.

Authors’ responses

Indeed, the small sample size is a limitation of the study, limiting the generalizability of the results. We mentioned this in the “strengths and limitations” sub-section.

3) There are many measures. However, the reliability and validity data are lacking.

Authors’ responses

All measures used were validated tools for this clinical condition, and covered different aspects of these patient’s characteristics to ensure the reliability of the study. However, no sampling methods were used and the sample size is small which is a limitation to the reliability of the study.

We mentioned this in the “strengths and limitations” sub-section.

4) How did the authors deal with multiple testing?

Authors’ responses

This pilot study did not adjust for multiplicity or use hierarchical methods. The aim is to provide information to formulate hypothesis for further studies. Statistical analyses were exploratory.

We mentioned this in the “analysis” sub-section and conclusion.

5) Did the authors address the severity of the ASD symptoms?

Authors’ responses

According to the ADOS 2 calibrated severity score, three patients had mild autistic symptoms, 6 had moderate symptoms and only 1 had severe symptoms.

We have mentioned this in the results section (subsection 3.2.). We have also included a paragraph on the calibrated ADOS-2 severity score in the manuscript (subsection 2.2.21.).

6) The results are clear and well-presented.

Authors’ responses

Thank you for your comment.

7) Please review Table 5. I think it should be moved to the results section.

Authors’ responses

We have moved table 5 to the results section.

8) What about the Conclusion section?

Authors’ responses

As suggested by the reviewer, we've added a conclusion section.

“Through a pilot study comparing the developmental profile of children with ASD versus those with SCD, some differences have been identified. In particular, it appears that in addition to the RRBs present in the ASD group and not in the SCD group, socialization skills are less impaired in the SCD group. Moreover, the total score on the ADOS-2 is lower in SCD, on average below the significant cut-off score for SCD.

While these initial results are interesting, it is important to replicate the study to confirm them with a larger number of children. In addition, it would enable us to compare cases of SCD with various forms of ASD depending on the level of intensity of the symptoms.”

Reviewer 2 Report

Comments and Suggestions for Authors

Thank you for the opportunity to review this manuscript.

The research refers to interesting populations, the goal of the research is clear and can expand the information available in the literature. However, I have some comments that may help to improve the manuscript.

1) The Introduction is well-written, and provides important information, the goals are clear. However, the authors should add study questions or / and hypotheses. In addition, it would be helpful to add headings in the introduction.

2) The sample size is small. This limits the possibility of generalizing the results.

3) There are many measures. However, the reliability and validity data are lacking.

4) How did the authors deal with multiple testing?

5) Did the authors address the severity of the ASD symptoms?

6) The results are clear and well-presented.

7) Please review Table 5. I think it should be moved to the results section.

8) What about the Conclusion section?

Author Response

1) The Introduction is well-written, and provides important information, the goals are clear. However, the authors should add study questions or / and hypotheses. In addition, it would be helpful to add headings in the introduction.

Authors’ responses

At the request of reviewer 2, we have added our hypotheses at the end of the introduction.

We have also added headings in the introduction.

2) The sample size is small. This limits the possibility of generalizing the results.

Authors’ responses

Indeed, the small sample size is a limitation of the study, limiting the generalizability of the results. We mentioned this in the “strengths and limitations” sub-section.

3) There are many measures. However, the reliability and validity data are lacking.

Authors’ responses

All measures used were validated tools for this clinical condition, and covered different aspects of these patient’s characteristics to ensure the reliability of the study. However, no sampling methods were used and the sample size is small which is a limitation to the reliability of the study.

We mentioned this in the “strengths and limitations” sub-section.

4) How did the authors deal with multiple testing?

Authors’ responses

This pilot study did not adjust for multiplicity or use hierarchical methods. The aim is to provide information to formulate hypothesis for further studies. Statistical analyses were exploratory.

We mentioned this in the “analysis” sub-section and conclusion.

5) Did the authors address the severity of the ASD symptoms?

Authors’ responses

According to the ADOS 2 calibrated severity score, three patients had mild autistic symptoms, 6 had moderate symptoms and only 1 had severe symptoms.

We have mentioned this in the results section (subsection 3.2.). We have also included a paragraph on the calibrated ADOS-2 severity score in the manuscript (subsection 2.2.21.).

6) The results are clear and well-presented.

Authors’ responses

Thank you for your comment.

7) Please review Table 5. I think it should be moved to the results section.

Authors’ responses

We have moved table 5 to the results section.

8) What about the Conclusion section?

Authors’ responses

As suggested by the reviewer, we've added a conclusion section.

“Through a pilot study comparing the developmental profile of children with ASD versus those with SCD, some differences have been identified. In particular, it appears that in addition to the RRBs present in the ASD group and not in the SCD group, socialization skills are less impaired in the SCD group. Moreover, the total score on the ADOS-2 is lower in SCD, on average below the significant cut-off score for SCD.

While these initial results are interesting, it is important to replicate the study to confirm them with a larger number of children. In addition, it would enable us to compare cases of SCD with various forms of ASD depending on the level of intensity of the symptoms.”